

**Water in the Critical Zone: Soil, Water and Life from Profile to Planet**
M.J. Kirkby
School of Geography, University of Leeds, UK.
m.j.kirkby@leeds.ac.uk
**Abstract**
Earth is unique in the combination of abundant liquid water, plate tectonics and life,
providing the broad context within which the critical zone exists, as the surface skin of the
land. Global differences in the availability of water provide a major control on the balance of
processes operating in the soil, allowing the development of environments as diverse as those
dominated by organic soils, by salty deserts or by deeply weathered lateritic profiles.  Within
the critical zone, despite the importance of water, the complexity of its relationships with the
soil material continue to provide many fundamental barriers to our improved understanding,
at the scales of pore, hillslope and landscape.  Water is also a vital resource for the survival of
increasing human populations.  Intensive agriculture first developed in semi-arid areas where
the availability of solar energy could be combined with irrigation water from more humid
areas, minimising the problems of weed control with primitive tillage techniques. Today the
challenge to feed the world requires improved, and perhaps novel ways to optimize the
combination of solar energy and water at a sustainable economic and environmental cost.

**1.   Water and critical zone typology**
The earth provides a unique planetary environment in which liquid water, plate tectonics and
life have co-evolved to create the critical zone. Our existence relies on the properties of this
dynamic soil layer and the ways in which water helps to maintain and regenerate the
ecosystem services that it provides.  Soil properties have been described (Jenny, 1941) as
depending on climate, biota, relief, parent material and time. Although not explicit in this list,
water plays a vital part in almost all soil processes, mediating their dependence on these five
factors. In figure 1, the factors most directly linked to soil development have been re-
arranged to show the central role played by water in soil processes (Hillel, 1971).  Although
climate, parent material and atmospheric exchanges may be regarded as the (semi-
)independent external controls on soil development, water plays a vital role as an
intermediary, especially between climate, biota and soil.  The close interdependence of all
these processes demands  multidisciplinary research (Brevik et al, 2015) to deepen our
understanding.
The availability of water depends on the climate, defining the amount of precipitation, its
form, seasonality and variability from year to year.   Water typically spends months (soil
water) to centuries (groundwater) within the critical zone, allowing it to interact effectively
with soil and bedrock constituents. The areal distribution and seasonal pattern of rainfall and
evapotranspiration are, therefore, perhaps the strongest global scale controls on critical zone
development.
Water in the soil provides an essential working fluid for plant growth, being directly involved
in photosynthesis and providing turgor, and is vital for all organisms in the soil (Bardgett et
al, 2001).  The way in which biota interact with and influence the critical zone is strongly
linked to the intensity of water circulation through living organisms.  Where water is freely
available, and potential evapotranspiration is high, biomass is generally high, including both





vegetation and soil organisms. Decaying vegetation provides soil organic matter and also
provides an important resource for the soil organisms that enhance decomposition, as well as
a dynamic reservoir for soil water.
Flowing water, wetting and drying, freezing and thawing all physically move soil aggregates
to transport the soil, progressively modifying the topography, and so the way in which the
critical zone interacts with relief.  As relief is progressively lowered, sediment transport is
generally less, due to the lower potential energy of overland and subsurface flow, whereas
chemical removal is much less affected by the slower water drainage.  This trend generally
leads to a deeper and more weathered critical zone, which progressively modifies the soil
structure and the pathways of water moving over and through the soil, and with organisms
actively exploiting the system to their advantage.  Within-slope effects are also observed as
sediment and organic matter is transferred from upslope to downslope sites, particularly
through tillage erosion (e.g. Wright et al, 1990, van Oost et al, 2005).
Water interacts with nutrients and weathering products, and its flow redistributes dissolved
material.  Water in the parent material acts as solute, dissolving weatherable minerals and
making them available for advective transport in flowing water and diffusive transport in
immobile water (Kirkby 1985).  In arid conditions, material dissolved from parent material or
deposited in the wind is often re-precipitated within the profile as, for example, sodium salts
or calcrete. In humid conditions, solutes are largely carried away, progressively weathering
the residual soil.
Over time, the critical zone progressively evolves over similar time spans to the evolution of
the entire landscape. In some shield areas, this process appears to continue for many tens of
millions of years, but, more commonly the critical zone approaches a near steady state of
almost constant mechanical and chemical denudation in which the structure and form of the
critical zone is only very slowly changing while the landscape is continuously lowered at a
steady rate (Riebe at al, 2003).  The critical zone is in a state of transient change until it
reaches a steady state, or in response to external shocks such as deforestation and climate
change, or the slow evolutionary changes in vegetation.  During such periods of transience,
the internal processes are strongly driven by changing soil hydrology.
At a global scale, the dominant control on soil development is the balance between climate
and atmospheric inputs.  Climate controls the overall soil hydrology, that can be expressed by
the relationship between precipitation and potential evapotranspiration.  Atmospheric inputs
or outputs are partially dependent on the climate.  Dust is perhaps the most important single
component, source areas being associated with little vegetation cover and at least some dry
periods when the surface material can be entrained. Desert areas are the most important
source areas, but current and former glacial outwash areas are also important, currently
generating about 10% of the global dust budget (Bullard, 2013).  Areas downwind of source
areas receive dust, which is widespread globally, but most concentrated close to source areas
due to selective transportation of silt-sized material. Particularly high concentrations form the
areas of extensive loess accumulation, for example in China, northern Europe and the
American Mid-west.  Other significant atmospheric exchanges are associated with transport
of inorganic salts that are most concentrated close to the ocean or exposed evaporate deposits
(themselves more prevalent in current or former arid areas).  The relative importance of
atmospheric inputs as agents of soil formation is strongly dependent on the hydrological
balance, between precipitation and evapotranspiration.  (FAO, 1961; Prentice et al, 1992) In
figure 2, the hydrological balance is compared with the atmospheric exchange balance to



define broad regimes of soil development.  Where the hydrological balance is very strongly positive (precipitation greater than potential evapotranspiration)  throughout the year, then organic material accumulates at the surface and persistent waterlogging creates anoxic conditions that minimise decomposition of organic matter, which accumulates as an organic soil.  With a less positive and/or more seasonal hydrological balance, the critical zone is dominated by loss of dissolved weathering products and, given sufficient time, develops a deeply weathered profile, often lateritic.  Once almost all nutrients have been leached from the upper layers of soil, plants may eventually become largely dependent on atmospheric inputs of nutrients dissolved in rainfall.

Under arid conditions, where the hydrological balance is negative, most of the precipitation that enters the soil is lost in evapotranspiration, re-depositing any products of chemical weathering within the soil, most frequently as calcrete layers, in some cases increased through inputs of wind-blown dust.  Extremely arid areas provide ideal conditions for deflation, and, at the extremes, tend towards a rocky desert from which all fines near the surface have been removed.  If, however, there is accumulation of salts, from the sea or from evaporites, then surfaces are instead dominated by salt accumulation and undergo rapid weathering as salts crystallise within the rock (Lavee et al, 1998; Howell, 2009).

Thus, at a global scale, water relations dominate the whole course of evolution of the critical zone.  Some of these gross differences are modified by the different age of soils, with more rapid re-cycling in areas of Pleistocene glaciation, and opportunities for the accumulation of weathering products throughout the Cainozoic in some low latitude shield areas.

Although there are many alternative ways of conceptualizing the relationships between water and soil, the development of the critical zone concept has perhaps done more than any other to transform the study of the soil and to emphasise its essentially multidisciplinary nature (Brantley et al, 2007; Lin 201; Anderson & Anderson, 2010: Anderson, 2012: Brevik et al, 2015).  Although, inevitably, some aspects of this re-focussing overlap with existing components of earth science, the establishment of  Critical Zone Observatories (CZOs), first in the United States (Anderson et al, 2008) and now internationally (Banwart et al, 2012), is doing much to foster new research and improve our understanding of how soil is related to the landscape at hillslope to global scales.

## 2.  Movement of water within the critical zone

At finer scales, the relationships between water and soil are in the domain of soil hydrology and soil physics. In both  of these fields, there are many questions that are not fully resolved and, because of their importance, a considerable literature.

At the finest scale, flow of water within the soil is most commonly described by the Richards (1931) equation, re-stating, for an unsaturated soil, Darcy's (1856) law that the rate of flow is proportional to the pressure gradient.  There are a number of challenges  in interpreting the Richards equation for a real soil.  The first difficulty is that, in these expressions, the hydraulic potential is a measure of the capillary tension exerted by films of water held between soil grains and aggregates, and that this tension depends not only on the moisture content, but also on the previous history of wetting and drying.  With measurements of the relationships between soil moisture and tension (Buckingham, 1907),  it is possible to solve these equations in simple cases, such as for saturated infiltration into an initially dry soil (Youngs, 1957), but more general solutions are elusive.  One useful approach has been to



focus on the infiltration process, to develop  expressions that were broadly consistent with
the Richards equation, but relied on fewer parameters (Philip, 1954, 1969; Green & Ampt,
153 1911).
However, the Darcy/Richards approach assumed that as flow passes through  the soil, there is
complete mixing between the flowing threads of water, and there are commonly substantial
deviations from this assumption, because pore sizes and shapes vary, so that water travels
faster through macropores, bypassing flow through the finer pores of the soil matrix (Beven
and Germann, 1982, 2013). Macropores are widespread, due to the contrast in pore sizes
between and within individual soil aggregates, as well as more extreme contrasts produced by
cracking in clay-rich soils and open pore spaces around stones in the soil.  In some cases the
behaviour of the soil can be dominated by flow in either the matrix or the macropores, but
this response varies with the moisture content as well as over time in swelling soils.  In many
case therefore a more complex model is required, for example involving dual porosity and
marked hysteresis.  Experimental evidence is showing the intricate three dimensional patterns
of wetting and draining in a block of soil (e.g. Weiler & Naef, 2003; Haber-Pohlmeier et al,
2009) but, to date, there is no simple model that adequately describes the range of observed
behaviours.  Simple infiltration equations are still being applied as a necessary
phenomenological tool, but it is clear that they can only represent a single prior soil state, for
example ponded infiltration into an initially dry soil.
Unsaturated flow in the soil takes place predominantly in the vertical direction, as rainfall
percolates toward a saturated level (if there is one) where lateral flow occurs, predominantly
in the saturated phase. This contrast reflects the lower hydraulic gradient and the much larger
distances involved in lateral flow, so that only saturated flow is able to drive significant
volumes of water.
Under climates, and during seasons, where precipitation is less than potential
evapotranspiration the movement of water is predominantly vertical: infiltrating water
supplies evapotranspiration, only penetrating deeply into the soil in the largest storms, and
there is little or no surplus to drive lateral flow. When precipitation exceeds potential
evapotranspiration, the excess can only be carried away by lateral flow, which may be
overland, within the soil or in groundwater. This dichotomy, often with seasonal switching
between these modes (Grayson et al, 1997),  shows strong contrasts in the downslope
connectivity that is established by lateral flow. In a place and season dominated by vertical
fluxes, each point responds independently to the rainfall supply and evapotranspiration
demand, and common responses are filtered by local heterogeneities. Lateral connectivity is
only briefly established during relatively infrequent flow events, usually overland, so that
behaviour at a point responds only to very local influences.  Where lateral flow is dominant,
there is a near-continual connection, commonly subsurface, and the hydrological response at
any point integrates the effects of every point upslope that drains towards it.
At the soil surface, overland flow is generated either when rainfall exceeds the infiltration
capacity (Horton, 1933) or when the surface soil is saturated (Hewlett & Hibbert, 1967;
Dunne & Leopold, 1975; Kirkby, 1978; Beven, 2000). The former, infiltration excess
overland flow, is dominant in semi-arid areas where rainfall exceeds potential
evapotranspiration so that the soil is dry. Rainfall impact crusts the bare soil surface around
the sparse vegetation, while shrubs may funnel water towards their roots (Cammeraat et al,
2010) setting up a strongly contrasting patchwork of infiltration.  The latter, saturation excess



overland flow, occurs mainlyin humid areas, where rainfall is greater, generating substantial
subsurface flow and a strong vegetation cover. However under many Mediterranean and
other seasonal climates, there is switching between these modes during the year and, even in
humid areas, extreme rainfalls may generate infiltration excess overland flow. When
saturated overland flow occurs, the contributing area commonly expands as saturation builds
outward from stream banks and stream-head hollows, driven by concentration of subsurface
flow from upslope and the accumulation of rainfall on the nearly saturated ground.
Infiltration excess overland flow, when it occurs, tends to be generated more uniformly, so
that flows, when they occur, tend to be more flashy and more damaging. However, there
remains a very strong variability in local infiltration capacity, so that, particularly at the
beginning of a storm, the detailed pattern of overland flow is characterised by patches of flow
generation and re-infiltration which persist until flow becomes general (Kirkby, 2014) and is
then dominated by local flow convergence steered by the micro-topography (figure 3).
In humid areas, particularly under forest, there is an extensive literature on subsurface flow
mechanisms. There appears (Tromp-van Meerveld and McDonnell, 2006) to be strong
similarity in many cases between the mechanisms of subsurface flow and those of infiltration
excess overland flow. In each case, rainfall fills depressions and/or infiltration storage and
flow begins as these progressively spill over to form connections. The surface for which this
process is most critical may be the ground surface (for infiltration excess overland flow), or a
subsurface level below which there is a sharp decrease in permeability, whether due to soil
layering or at the soil-bedrock interface. Experimental (e.g Graham et al, 2010) and
modelling data (e.g Kirkby 2014, Penuela et al, 2015) supports percolation theory (e.g.
Wikipedia 2016) in finding that the response of such a system to increasing rainfall amounts
shows a rather sharp threshold, below which there is negligible flow, and above which there
is transition to a near-linear increase in connected flow. Since the sharpness of the threshold
varies, it may be best to define the storm rainfall at which there is a 50% runoff as an
operational threshold.
Two valuable ways of generalising response at the hillslope scale are through the concepts of
connectivity (McGuire & McDonnell, 2010: Bracken et al, 2015) and residence time (e.g.
Tetzlaff et al, 2010). At its simplest, connectivity queries the presence or absence of a
through connection between two points. However, it has proved more fruitful for describing
connections across an area, and is thereby linked to the runoff coefficient. Connectivity has
been widely applied in ecology (McCrae et al, 2008) applying an analogy with electrical
conductivity, but the one-way nature of water flow downhill makes this less applicable in
hydrology. Instead the application of percolation theory or the concept of a breakthrough
volume to establish connections have proved more applicable, and continue to be developed.
Residence time is, in a way, the inverse of connectivity, long residence times being
associated with poor connectivity and vice-versa. The great value of residence time is that its
mean value and distribution can be quantified using tracer methods. Perhaps these methods
may provide the basis for a better understanding of how water interacts with the critical zone,
by focussing on the hillslope rather than the soil profile scale,
Some storm precipitation is not involved in this fill and spill process. Until break-through
occurs, all of the rainfall; and after break-through a small fraction, percolates downwards
commonly reaching a level of saturation. Where and when precipitation is of the same order
as, or exceeds potential evapotranspiration, this downward percolation contributes to lateral
subsurface flow that brings the saturation level progressively closer to the surface in response



to flow rates that respond to hillslope plan and profile form: in less humid areas this
percolation only occurs in the largest storms, and most water is lost to evapotranspiration.
Subsurface flow between and during storms, if it occurs, establishes a dynamically varying
saturated area, usually along slope-base concavities and plan-convergent stream heads.
Rainfall falling on these saturated areas cannot enter the soil, but is immediately diverted as
saturation excess overland flow.  The fill and spill level and the saturated subsurface flow
level may be vertically adjacent, distinct, or in multiple layers.  In many cases one or other of
these mechanisms dominates the hydrological response of a hillslope or headwater area
(Beven, 2000; Tarboton, 2003).   Both fill and spill mechanisms and saturated contributing
area mechanisms share a very strong non-linearity in response to storm size, corresponding to
the increasing connectivity of flow.  At extremes which are rarely achieved, there is 100%
connectivity, but most observations reflect the region of increasing partial connection (e.g.
Bracken at al, 2013),although the mechanisms of establishing connected flow differ greatly.
For infiltration excess overland flow and other fill and spill regimes, connection is typically
established dynamically during the course of each individual storm, and decays rapidly
afterwards.  For saturation excess regimes,  initial connections are established by subsurface
flow that persists between storms in areas where precipitation exceeds potential
evapotranspiration.  The saturated area continues to expand during a storm, and connectivity
is only slowly lost, often on a seasonal time scale (Reaney et, 2014).  Over a period, an area
may experience fill and spill runoff when storm rainfall exceeds some threshold; and may, at
other times, experience saturation excess runoff when net rainfall over a period exceeds
another threshold.  The fill and spill threshold depends on the structure of vertical storage
within the soil, whereas the saturation excess threshold depends on topographic wetness
index and near-surface lateral permeability (Sivaplan et al, 1987, Kirkby et al, 2008).  Clearly
semi-arid areas, with little net rainfall, rarely experience saturation excess runoff, but both
mechanisms can co-exist in an area, often with seasonal switching between the two modes
(Kirkby et al, 2011)
**3.  Water as a transporting medium in the critical zone**
As parent material weathers, breaks down physically and is eventually removed by erosion at
the surface, it passes through the critical zone from bottom to top.  Figure 4 sketches the path
of grains from the parent material for a steady state in which the critical zone depth remains
constant, surface erosion balancing advance of the weathering front.  Initially a grain is
subjected mainly to chemical weathering, so that it approaches the surface vertically, relative
to the downward advancing weathering front.  As grains get closer to the surface, they
become increasingly influenced by diffusive movements in the soil,. These all gradually
move material down-slope, at a rate that decreases with depth.  Eventually erosion will
expose grains at the surface and remove them (Anderson et al, 2002).  There is a lateral flux
of eroded sediment and weathering products in solution at every point down the length of the
hillslope, and, perhaps after intermediate deposition, this material is finally exported at its
base, normally to a channel.
Water plays an essential part in these processes, generally percolating less and less with depth
where it interacts with rock minerals to release solutes and advance the weathering front
(Anderson et al, 2007;  Kirkby, 2015). The water is then partly diverted laterally, and partly
returned to the surface as evapotranspiration. In semi-arid climates, where potential
evapotranspiration exceeds precipitation, there is little lateral movement, and many solutes
are re-deposited beneath the surface. In more humid climates, lateral flow carries solutes
away, and weathering produces a much greater loss of rock substance.





Within the soil, slow diffusive movement is commonly driven by freeze-thaw, wetting-drying
and/or bioturbation, and all of these respond positively to the presence of soil water.  Where
the slope configuration is suitable, larger and more rapid mass movements can also move
critical zone materials downhill, usually under conditions close to saturation.  At the surface,
raindrop impact and overland flow drive soil erosion, which is most effective where the
surface is not protected by vegetation or stone cover.  In all of these ways, the action of water
is strongly instrumental in shaping the path followed by grains as they migrate through the
critical zone (figure 4) and progressively reduce in grain size.
Weathering processes progressively convert strong rock minerals, that have generally been
synthesised under high temperatures and pressures in an anoxic environment, to weathering
products that are closer to equilibrium with surface conditions and oxygen levels. In this
process most minerals  lose strength, eventually converted to granular sand or silt and  clay
minerals.   This loss of strength and reduction in grain size facilitates lateral movement of
weathered material close to the soil surface.  The balance between chemical(C) and
mechanical (M) denudation rates determines the degree of weathering of surface soils. The
depletion ratio (Riebe et al, 2003), defined as $C/(C+M)$ is a measure of the degree of
weathering in the soil, and generally increases in humid climates (with high C) and decreases
where slope gradients are high (with high M). Water circulation is progressively reduced with
depth in the soil. Low rates of mechanical denudation reduce stripping of the soil, which then
accumulates to greater depth and, in turn,  reduces chemical denudation, so that depletion
ratios, in any given rock/climate environment, tend towards a stable end-point value.
Water also plays an important role in the carbon and nitrogen cycles that are central to
biological activity. Carbon is fixed in photosynthesis from $CO_2$ in the atmosphere to
synthesise the carbohydrates that form the bulk of above and below ground plant tissue. It is
released as surface litter above ground, and by root decay within the upper parts of the critical
zone, where it accumulates as soil organic matter that gradually decomposes to release $CO_2$
first back into the soil air and eventually to the atmosphere. Saturation with water, by
reducing oxygen in the soil, greatly slows decomposition, changing the environment and
effectiveness of the soil microorganisms responsible.  Except in the most arid conditions, the
rate of oxic decomposition increases with temperature.  Soil organic matter is very effective
in readily storing and releasing soil water, thereby acting as one critical reservoir for plant
water uptake.  Soil organic matter also provides nutrition for worms, termites and other
macro-invertebrates in the soil that physically mix and aerate the soil, accelerating both
decomposition and water exchange.
Nitrogen is fixed from the atmosphere, mainly by fungi, strongly enhancing the ability of
plants to synthesise the proteins which are essential for healthy growth. These macroscopic
and microscopic organisms rely on soil organic matter and the water in it for their own
metabolism.  The vegetation, carbon and nitrogen cycles therefore mutually reinforce each
other, relying on water as a key medium for the uptake of nutrients.  Some organic nitrogen is
incorporated into soil organic matter, and some breaks down, mostly to nitrate which is
highly soluble in water and so is readily lost in runoff.
Much of the Nitrogen in circulation comes from the application of artificial fertiliser for
agriculture. Although essential for the increased yields it promotes, it is also one of many
organic and inorganic pollutants that reach the soil through direct application and/or in wet
and dry deposition (Keesstra et al, 2012). These pollutants or their metabolites play an



increasing role in contaminating stream groundwater, with potentially adverse effects on soil
organisms and human health.
Overland flow, however generated, is the key agent of soil erosion.  Unprotected soil surfaces
are impacted by raindrops that break up and detach surface aggregates, packing some down
to crust and seal the surface, and ejecting some either into the air (rainsplash) or, where water
is already flowing downslope, into the flow (rainflow). Once flow becomes sufficiently
strong, due to topographic convergence and/or at high rainfall intensities, the tractive stress
exerted directly by the flowing water becomes sufficient to erode the surface and detach
material (rillwash).  When this happens, the flow begins to incise channels into the surface,
thereby increasing the convergence of flow lines in a positive feedback that leads to rilling or
gullying.  All of these processes are highly size-selective, transporting the finest material
farthest from its detachment point, and rates of movement increase with slope gradient.
Surfaces may be protected either by vegetation or by stone cover.  The crown cover of
vegetation breaks the impact of falling raindrops, so that they then strike the ground with low
momentum and detach little material (except under high-crowned trees).  Stones protect the
surface directly, and each stone tends to shield a rim of  soil in its immediate shadow, so that
it does not become crusted (Poesen et al, 1994; Cerda, 2001).  Crusting, particularly in silt-
rich soils, is very effective in reducing infiltration and therefore increasing overland flow and,
indirectly, erosion. Where the soil is stony, initial erosion tends to winnow out the fine
material until the stones, that are less easily carried way, are left behind to armour and
partially protect the surface from further erosion.  Deep gullying is therefore strongly
associated with deep soils that are deficient in stones, either through the action of weathering
or as a property of the parent material: thick loess deposits provide an extreme example.
Agricultural fields, at times of year when the surface is almost bare, are generally vulnerable
to greater erosion than areas of semi-natural vegetation, particularly so when this period is
also one with a high risk of intense rainstorms.  During a severe storm, rills generally form
with a more or less regular spacing. At the same time as their bed is being incised by the
concentrated flow, material is also being delivered to them by rainsplash and rainflow from
the intervening areas, so that their downward incision may be self limiting, often cutting
down only to a hardened plough-pan level.
On all but the steepest slopes, slow mass movements, soil erosion by water and chemical
denudation are the dominant processes through which hillslopes evolve over time.  Under
diffusive processes such as soil creep, rainsplash and tillage erosion, hillslope profiles
gradually evolve towards a mainly convex form, with a narrower concavity towards the base,
as has long been observed (Gilbert, 1877).   In steeplands however, rapid mass movements
assume the dominant role, and tend to produce almost rectilinear slopes once cliffs have been
eliminated.  The much higher rates of sediment transport create a critical zone that generally
remains thinner, and with lower chemical depletion ratios than on lower-gradient slopes, even
though the shallow soil depth promotes a relatively high rate of chemical denudation
(Emberson et al, 2015).
Sediment transport, also shapes the three-dimensional landscape geometry, through the
interplay of diffusive and advective sediment transport processes.  Where advective sediment
transport by water is able to evacuate sediment faster than it can be replaced by diffusive
processes or mass movement, then channels become progressively incised, defining the



drainage density of the landscape, and so the average length of hillslope profiles. This is a
dynamic process in which major storms are responsible for headward stream extension, and
fresh headcuts are partially infilled between major storms, so that the instantaneous stream
head position fluctuates, reflecting recent storm history. Drainage density tends to be higher
in more arid climates, reflecting the dominance of surface flow processes where vegetation is
sparse. Density also tends to increase with valley gradient, because advective transport
generally increases more than diffusive transport as gradient increases (Montgomery &
Dietrich, 1992).

The interplay of hillslope and channel processes responds not only to climatic variablility but
also to land use changes that modify sediment supply, most strikingly following changes in
land use. Where land use change exposes more bare soil, as in deforestation and adoption of
arable farming, runoff and sediment load tends to increase. Channel runoff is generally less
strongly affected by local changes, so that the increased sediment delivered from side slopes
is redeposited along channelways because their transporting capacity is not proportionally
increased (Rommens et al, 2005). Contrariwise, afforestation can lead to stream incision
(Keesstra, 2007, Sanjuan et al, 2010).

These considerations show that water plays a crucial role in almost all processes acting within
the critical zone, and across the full range of landscape scales (Brantley et al, 2007; Anderson
et al, 2015). Although other factors, such as lithology and tectonics, also play a very
important role, climate, principally acting through the availability and distribution of water,
has a dominant influence on the structure and composition of the soil, on the rates and styles
of mechanical and chemical denudation, and on the profile form, plan shape and length of
hillslope profiles. Many of the processes involved in shaping three-dimensional hillslope
form are now being incorporated into successful landscape evolution models (Tucker et al,
2001; Egholm et al, 2013) including the effects of non-linear diffusion (Roehring et al,2001).
However, the incorporation of chemical solution in these models perhaps remains their least
satisfactory component (Brantley et al, 2007).

**Water for plant growth**
Roots grow actively to seek pore water which they require to maintain their turgor against
strong capillary tension and to permit photosynthesis. Except where a saturated zone in
within reach of their root system, the water that plants use appears to come mainly from the
matrix within soil peds, and is substantially separate from the water in cracks and macropores
between aggregates that is the main contributor to stream flow (McDonnell 2014). When
water flows through macropores, it also infiltrates into the peds beside each macropore,
recharging the soil matrix which then provides a longer lasting store of water to supply the
plants (Germann and Beven, 1985). Water also supports soil microflora, especially fungi,
bacteria and viruses; and fauna from termites and earthworms to nematodes and protozoa that
graze, mainly on bacteria and living plant material or their organic matter residues. Bacteria
are important in catalysing weathering processes and some fungi (mycorrhiza) support plant
growth by fixing atmospheric nitrogen. The various soil organisms are essential to a healthy
soil, and may contribute up to 10% of the total biomass.

It has been argued (Schymanski et al, 2008) that the vegetation cover develops in such a way
as to maximise its productivity, and such a principle of optimality may be a way to simplify
the complex web of interactions linking vegetation and soil organisms to water use. Most
existing models, however, use a more physically based set of constraints to model vegetation
and how it may respond to global climate change (e.g. Scheiter et al, 2013).



Matrix water is most abundant near the soil surface, since macropores are most frequent there, and are commonly active with every rainfall event. Root distributions tend to mirror this distribution, often with a more or less exponential decay in density with depth. Some plants also develop deep tap roots that can reach down to a water table at 10 m or greater depths, a strategy favoured by semi-arid phreatophytes that exploit local water tables below ephemeral streams.

Although water is not directly responsible for the structure and processes within the root zone, its presence and distribution, acting through the vegetation and soil organisms, enables the processes of decomposition and bioturbation that dominate these surface layers of the critical zone, and these processes profoundly modify the soil structure and hydrology.

Plant roots and mesofauna (e.g. earthworms and termites) physically break up the soil, allowing the penetration of air and water. Larger burrowing animals, falling trees and freeze-thaw or wetting-drying cycles can also play a part in breaking up and mixing the near-surface soil. The cumulative action of all these processes can be considered as a diffusive mixing, with a net upward drift towards the free surface, which is counterbalanced by settling under gravity, significantly assisted by the downward percolation of water (Gabet, et al, 2003). Over a few decades, the balance between these processes leads to an equilibrium bulk density profile, in which porosity declines with depth. This bioturbation mixes and homogenizes the upper layers of mineral soil, since it occurs much more rapidly than chemical weathering, and may readily be visually distinguished from weathered saprolite, in which original bedrock morphology is preserved in the weathered material.

Organic matter is released from plants, partly as leaf (and stem) fall to accumulate on the surface and partly as in-situ root decay. Over decades, this material takes part in the vertical mixing and also decomposes, gradually releasing $CO_2$ into the soil. Since the processes of mixing and decomposition occur over similar time-spans, the soil organic matter also develops a vertical distribution within the soil, generally with a smaller scale depth of exponential decay than for bulk density.

These mixing processes, by modifying the near-surface soil, tend to increase the rate at which water is able to infiltrate, creating a positive feedback in which greater biological activity increases the availability of water in the soil, which in turn encourages biological activity. Eventually, soils are able to absorb the available precipitation so that, over a time span of decades to centuries, there is a tendency for soil structure increasingly to reflect the natural vegetation and to reduce overland flow runoff.

## 4. Water as an agricultural and food resource

By far the greatest use of water by mankind is for agriculture. An average of approximately 3,800 litres a day is needed to support each individual (Hoekstra & Mekonnen, 2012), 92% of which grows their food. Other major requirements are for domestic use (3.8%) and for clothing and other industrial products (4.7%). These requirements differ in kind, in that domestic water has to be delivered to the individual, whereas for other uses the water can be more economically provided by transporting the food or clothing. However, most countries are also concerned with food security, so that there is some perceived pressure towards being at least partially self-supporting for food production.



499 Historically, the development of large scale agriculture has been in semi-arid regions of the
500 Middle East and Meso-America, commonly using irrigation water canalised from rivers
501 (Mazoyer & Roudart, 2006). Semi-arid areas have the advantages of providing ample solar
502 energy for photosynthesis, together with relative ease of weed removal. However, irrigation
503 has, historically, commonly led to salinization of the soil, sometimes irreversible, depending
504 on the quality of the irrigation water and whether sufficient irrigation water has been applied
505 to leach excess salts..  Clearance of land in warm humid regions, although providing ample
506 water and solar energy, is hampered by the re-establishment of native weed species and rapid
507 leaching of topsoil nutrients.  Long fallow periods (shifting agriculture) were therefore
508 required until modern machinery and fertilisers could be applied.  Increasing population
509 pressure may also place pressure on land resources, forcing undesirable reductions in the
510 fallow rotation period. In all areas, seasonal exposure of the bare land surface prior to
511 planting and after harvest, expose the land to increased soil erosion, particularly when rainfall
512 is intense at these critical times of year.  In the great majority of cases, arable farming
513 increases the natural rates of soil erosion by water, increasing losses by at least an order of
514 magnitude (Montgomery, 2007) and progressively degrading the land.  Water erosion takes
515 some steeper, thin-soil areas out of production and, more widely, removes the most nutritious
516 topsoil and organic material.  Cultivation, by exposing the soil surface and allowing it to dry
517 out, can also increase wind erosion in semi-arid areas (e.g. Houyou et al, 2014).  In addition,
518 conventional ploughing, whether on the contour or downslope, moves material downslope
519 and generally exposes soil organic matter to more rapid decomposition, reducing the long-
520 term water holding capacity of the soil. However, because significant deterioration of the soil
521 takes many decades, farmers may have little short-term incentive to improve conservation
522 practices.
523
524 Some of the negative effects of agriculture can be mitigated by appropriate management (e.g.
525 Keesstra et al, 2016), but these often require initial and ongoing investment that is not
526 available to all farmers. Some soil conservation measures such as inter-cropping can be
527 applied at low cost but the majority, including terracing, contour ploughing, residue
528 management, water harvesting, and reduced tillage, require investment and/or some sacrifice
529 of cultivable land area.  Management systems that retain a vegetation cover reduce the loss of
530 sediment (Abrahams et al, 1994; Zhao et al, 2016) and organic matter (Gao et al, 2016), even
531 where runoff is not reduced.
532
533
534
535 As well as on-site management of water resources, there is a global shortage of renewable
536 water resource in the face of increasing populations.  There are a number of technical
537 solutions to these problems, for example breeding crops that require less water, irrigating
538 crops as efficiently as possible without incurring the risk of salinization and water harvesting.
539 Others, for example large scale desalinisation of sea water, carry significant costs that cannot
540 readily be accepted by increasing the cost of food to consumers.
541
542 The scale of the water shortage can be seen through global patterns of malnutrition.  Figure 5,
543 compiled by FAO (FAO et al, 2015) shows the proportions of national populations with
544 inadequate nutrition.  While issues of governance and local conflicts play a significant part in
545 this distribution, there is a strong underlying message about agricultural productivity and
546 availability of water, in that many of these areas are affected by shortage of renewable water
547 resources.



The broader implications, for global water needs and for food security, are analysed further in
figures 6-8 on a country by country basis, using data drawn from FAO reports (2000, 2016).
In figure 6, the population that can be supported by renewable water and potential arable land
resources is calculated, assuming that approximately 600mm of water per year is required to
grow a crop, so that water is the limiting factor in some areas, and available land in others.
Although this makes the optimistic assumption that water is freely transferrable within a
country, it can be seen that there remains a substantial shortfall in many countries, and that in
many areas the shortfall is due to lack of sufficient water resources, including for the two
largest populations in India and China.  Some continuing increase in world population,
together with further deterioration of soil resources due to erosion and salinization, therefore
presents a major challenge for the future.
Figure 7 shows the actual and potential arable land in each country and the average
renewable water in each.  The horizontal line is set at 600mm, which is the approximate
amount of water required to grow a cereal crop.  It can be seen that many countries have not
sufficient water to make full use of their presently utilised arable land resources, and that
water limitations are a major factor in preventing cultivation of additional potential arable
land.  Figure 8 shows the renewable water resources per capita for each country, plotted
against the population.  The upper horizontal line shows the approximate amount of water
required to grow food for the population (ca 1200 $m^3$ per year), and the lower line the amount
needed for domestic use (ca 60 $m^3$ per year).  It can be seen that there are many countries that
cannot feed themselves, and a smaller number that lack sufficient renewable water to supply
domestic needs.  Although food needs can be and currently are being partially met by
international trade, with implicit water transfers within the food, lack of food security
remains a source of potential conflict.
One renewable and low cost means of increasing available water is through water harvesting.
Where rainfall is almost adequate for rain-fed farming, conservation measures may be all that
is needed to ensure that storm runoff is retained on-site, in mulch layers, in trenches or behind
bunds.  As water scarcity increases, an area to be cultivated can be supplied with runoff from
a collecting area above and around it, which provides water to the cultivated patch.  The
required ratio of cultivated area to collecting area can be estimated as the ratio of actual to
potential evapotranspiration in a given region.  Analysis of the climate thus gives some idea
of where different styles of water harvesting can be applied most effectively.  Figure 9 shows,
for Africa, the ratio of actual to potential evapotranspiration for the most suitable 5-month
growing season, and for the worst 25% of annual conditions.  Water harvesting can benefit
crop yields when this ratio lies between about 0.2 up to about 1.5.  The ratios of collecting
area to irrigated area are the reciprocals of these values, and are generally somewhat larger
due to the inefficiencies of collection and redistribution. Where the collecting area is large,
runoff collection can be made more efficient by removing stones from the surface  to
encourage crusting of the soil, and by channelling runoff into discrete runnels to further
reduce infiltration losses.
Where the collecting: cropping area ratio is low (<3), an individual group of plants can be
supplied by an immediately adjacent patch of bare soil.  This gives a pattern of pits where
seeds are planted, each surrounded by a small area that drains towards it (zai).  At larger
scales, a small planted field of, say 10x10 m, may be supplied with water collected from a
small upslope min-catchment (jessour), with perhaps a 20:1ratio of collecting area to irrigated
area.  At the coarsest scale, these practices merge into regional irrigation systems. Increases
in the ratio of collecting area to cultivated area lead to increased yields, but this may be offset




by the sacrifice of potential yield from the uncropped area where this is also suitable for
arable farming. Part of the net advantage is therefore obtained through greater labour
efficiency in farming a smaller area, and the reduced likelihood of crop failure. Figure 10
shows the modelled frequency distribution of estimated yields over a run of years, for Mekele
(northern Ethiopia), with different ratios of collecting to cropping area (CAR). It can be seen
that although most years provide some harvest, the median yield and its reliability are greatly
enhanced by water harvesting (Fleskens et al, 2016). Reliability can be further increased by
installing ponds that not only collect but also store water, often allowing irrigation in dry
spells, but subject to evaporative loss.
It is clear that water is a critical resource for agriculture, and will become more so in the
future (Falkenberg et al, 2009) , particularly because global warming is thought to increase
aridity in many water stressed areas (e.g. Gao and Giorgi, 2008 for the Mediterranean), and
because of the interactions with energy production (Pimentel at al, 2004). Figure 11 sketches
some of the interactions that need to be managed in order to maintain affordable food
production for a still growing world population. Crop production requires both water and
energy, mainly used for the manufacture of nitrogen fertiliser, but also for transportation on
the farm and to the market. Widespread use of fertiliser has quadrupled yields since 1900,
reducing the dependence of land as a critical resource, but without reducing the need for
additional water to support the increased crop yields, so that water has become the most
critical resource for further expansion of food production. Nitrogen fertiliser production uses
3-5 % of world natural gas as a source of hydrogen, and only a fraction of the nitrogen is used
by crops, so that nitrogen is a major source of pollution both directly in runoff, re-cycled
through animal manure, and as a greenhouse gas. Nitrogen in runoff contaminates
groundwater and is responsible for eutrophication of lakes and coastal waters.
Although water can be desalinated, the cost of irrigation water produced from seawater its
cost is very high in relation to other costs of food production. Typical developed world farm-
gate current cereal costs of about 200 USD per tonne (Zimmer, 2012) would be increased by
about 1000 USD per tonne for the desalinated water needed to grow the cereal (ca 1 USD per
cubic metre). These costs may be becoming acceptable for domestic water supply and for
some high value crops, but cannot, at present, be accepted for staple foods or animal
husbandry. Although desalination might be supported by renewable energy, for example to
irrigate the Sahara, it also generates disposal problems for the salt removed, and so is not
environmentally neutral.
Other pressures on water and land resources are also exacerbated as demand for food
increases, even more so if more meat is included in typical diets. Population growth will add
to existing pressure on cities, as agriculture is made more intensive. Urban and highway
development has already covered 2.3% of the European land area, much of it at the expense
of prime arable land. Agriculture has always been a risk factor, significantly increasing soil
erosion (Montgomery, 2007) above background rates. As more land is taken into cultivation,
it generally becomes more marginal, and so further increases the impact of erosion.
Irrigation, particularly where water is scarce, increases the loss of land to salinization.
Sealing, erosion and salinization all lead to some irreversible loss of cultivable land.
**5. Conclusion**
Water is everywhere. Life and mankind would not exist without it. As population continues
to grow, fresh water is becoming an increasingly scarce resource. To make the best use of
fresh water, most critically for food production, it is vital to share it wisely. One key aspect





of this is to progressively improve our knowledge of how water interacts with the critical
zone at every time and space scale, and to better recognise, and gradually stretch the limits of
what is possible. Water and soil present challenges at every scale, from the grain to the
globe, and it is a matter of urgency to engage with these issues as best we can, both as
practical problems requiring urgent solution and to enhance scientific understanding.

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



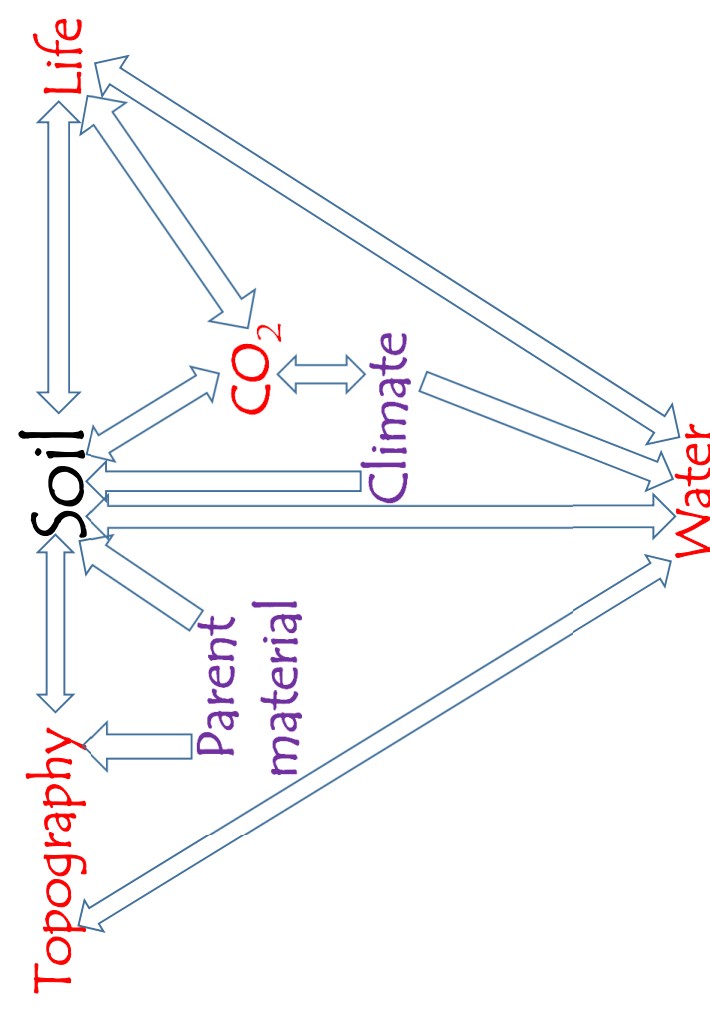

Figure 1

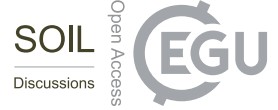

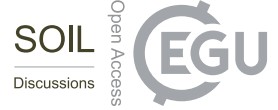

Figure 2



Figure3a





Figure 3b





Figure 4





Figure 5



Figure 6





Figure 7


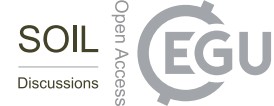

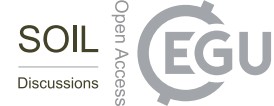

Figure 8

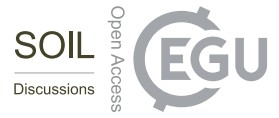

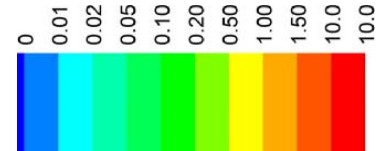

Water Harvesting potential, referred to worst 25% of annual conditions. Values 0.2 to 1.5 can benefit most from Harvesting.

Figure 9

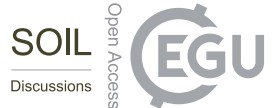

Figure 10





Figure 11