# Peer review of "Water in the Critical Zone: Soil, Water and Life from Profile to Planet"

_SOIL, 2016_

## Referee Comment (RC1) · Anonymous Referee #1 · 12 Oct 2016

This manuscript is interesting as it brings together several concepts from soils, hydrology , geomorphology and food production within the modern framework of the critical zone. This is the novel idea in the manuscript and as such worthwhile publishing The arrangement and structure of the different elements / paragraphs in the manuscript is very logical. However several of the subsection contain very general statements (section 1, 2 & 3) that are well known to the audience of Soil and I would suggest to condense these sections where possible. Several good textbooks exist on parts of the described interactions (e.g. Earth System Science by Jacobson et al. just to mention one) Sometimes also statements would need a reference or need some additional explanation. I missed in the first sections (sect 1) the human impact on soils, as mostly the natural conditions are described. This is addressed later in the paper but human impacts affects huge areas of our planet with regard to soil depth and quality. Maybe

in the introduction also a link could be made with recent publications on the Global sustainability goals as well as the concept of planetary boundaries (Rockwell 2009; Steffen 2015) in relation to the topic of the manuscript.

Specific comments l. 43: perhaps can be omitted here l. 55: soil aggregates: also individual grains are moved l. 60 (and 172): in low relief areas vertical movement of soil water (unsaturated zone) and leached elements are also key in the development of profile characteristics and horizons, which in the end determine many properties of the soil and its ability to sustain food production or ecological functions. l. 120: this paragraph could be written down more clearly and more explanation l. 213: explain figures 3a and b better in the text l. 227: suggest reference to Ali et al. HP, 2013 l. 347: in many densely populated regions the atmospheric deposition of N is responsible for an important increase in productivity in soil in natural environments l. 443: ref needed l. 482: maybe some other effects of plants could mentioned as well: organic acids produced by plants also play an important role in soil formation and movement of leached substances through the soil as well as natural acidification and nutrient depletion, that clearly affect biomass production l. 507: leaching: a would use the word depletion here l. 522: ref. needed l. 563: actually the limit for cereal growth is 300mm of rain but this is at the current open global market far from profitable (with an production of 0.5-1 ton.ha-1 at 300 mm) l. 568: I presume you mean 1200 and 60 m3 per year per capita l. 585: ref for the method applied? l. 595: reference on zai and jessour methods would be welcome l. 615 and l. 625: these two statements have been given earlier in the text l. 628: actually the cost is (or was) a bit lower, about 65 eurocent (Spain, Israel) see eg: Oñate and Peco, 2005 Figure3: explain what the color shades mean in the map: flow accumulation or infiltration rate , unit, and at what spatial scale In relation to figure 4 I would like to suggest a connection to the conceptual model of Stallard (1985) on the interaction between chemical weathering, soil depth and downslope transport in relation to topography Fig 6 and 7: can you give the data source?

Technical comments: l. 201: mainly in (typo) l. 287: soil,. (typo) l. 505: salts .. (typo) l. 788: reference is not listed corrected

---

## Referee Comment (RC2) · G. Govers (Referee) · 22 Oct 2016

Review of 'Water in the Critical Zone: Soil, Water and Life from Profile to Planet' by Mike Kirkby

In this paper, Mike Kirkby provides a comprehensive overview of the role of water in soils. This review paper is a very valuable contribution as it will serve as a resource for researchers that are not so familiar with soil science to understand the complexity of the subject and to quickly find which literature is relevant to deepen their knowledge. The last section adds to this an important perspective on why understanding soil water is important from a more applied viewpoint. In my opinion this paper can be published but I do think its impact can be improved with some modifications. There are a lot of suggestions in the annotated pdf. The most important issues are to me:

[Figure]

- The discussion on how water moves through a soil is interesting, but a key point is, in my view, neglected and that is that we often observe 'old' water to be pushed out during an event, rather than rainwater being spilled. It is, in my view, important that this concept is included and discussed in this review as otherwhise a too simplistic view may result (and remain imprinted in the reader's brain ;-). The work by Kirchner, Tetzlaff and others provides a very useful starting point to discuss this here. - There is confusion on the role of bacteria and funghi. Funghi form mycorrhizae (and are symbiotic with plants): they provide the plant with enhanced access to water (and dissolved minerals such as P) in return for carbohydrates (sugars) produced by the plant through photosynthesis. Several kinds of bacteria, on the other hand, can live symbiotically with plants as nitrogen fixers (exchange of reactive nitrogen in return for carbohydrates). This needs to be clarified. - I am not convinced that Figure 5 and the discussion that goes with it are necessary: I do think the reasons for malnourishment are, most certainly, not only water shortage and I think the discussion would be made clearer (and more relevant) by removing this figure. - I think the discussion of crop production and water is fascinating but that a key element is missing. Nowhere there is a mention of crop yields: I assume the author assumes that we need ca. 0.5 ha of cropland per person but this crucially depends on yields ! In Europe, such a surface area is sufficient to feed at least 5 persons, while it may not be enough in the Sahel. The main reason for this are the much lower yields in the Sahel, which are due to a lack of ag technology (Mueller et al., 2012; Neumann et al., 2010) rather than water alone. The same is true for water: water use efficiencies vary greatly around the globe (Rockström et al., 2007). This really needs to be accounted for and brought into the discussion. This does not mean that water is not a critical issue: it definitely is, but the discussion is too simplistic if we assume that all cropland produces (or can produce) the same. As possible analysis could start from actual yields and cropland areas and water efficiencies per country (available at FAO) and then make the calculations.

Gerard Govers

Mueller, N.D., Gerber, J.S., Johnston, M., Ray, D.K., Ramankutty, N., Foley, J.A., 2012. Closing yield gaps through nutrient and water management. Nature 490, 254-257. Neumann, K., Verburg, P.H., Stehfest, E., Müller, C., 2010. The yield gap of global grain production: A spatial analysis. Agricultural Systems 103, 316-326. Rockström, J., Lannerstad, M., Falkenmark, M., 2007. Assessing the water challenge of a new green revolution in developing countries. Proceedings of the National Academy of Sciences 104, 6253-6260.

-

Please also note the supplement to this comment:
http://www.soil-discuss.net/soil-2016-50/soil-2016-50-RC2-supplement.pdf

[Figure]

**Supplement:**

[revised manuscript text omitted]

Topography ⟷ Soil ⟷ Life

Parent material

$CO_2$

Climate

Water

[Figure]

Figure 2

[Figure]

[Figure]

Figure 3b

[Figure]

Lowering of surface by erosion

Penetration of weathering front

Paths of parent material through the critical zone

Export of sediment

Export of solutes

Figure 4

**PREVALENCE OF UNDERNOURISHMENT IN THE POPULATION**

**(PERCENT) IN 2014-16**

[Figure]

LEGEND

- <5% Very low
- 5% ⟶ 14.9% - Moderately low
- 15% ⟶ 24.9% - Moderately high
- 25% ⟶ 34.9% - High
- 35% and over - Very high
- Missing or insufficient data

Figure 5

[Figure]

Figure 6

[Figure]

[Figure]

Figure 8

[Figure]

Water Harvesting potential, referred to worst 25% of annual conditions. Values 0.2 to 1.5 can benefit most from Harvesting.

Figure 9

[Figure]

[Figure]

Figure 11

---

## Referee Comment (RC3) · Anonymous Referee #3 · 23 Oct 2016

In view of Mike Kirkby's five decades of leadership in hillslope hydrology and critical zone science (well before the term even existed), he has clearly earned the right to have his say on this subject. This wide-ranging review and synthesis will provide a useful introduction to the topic, particularly for graduate students and those new to the field, but experts of long standing will also find food for thought. In my reading through the manuscript there were many places where I said to myself, "but what about...?", only to find that the manuscript addressed exactly that point in the next paragraph or the next page.

Another reviewer has pointed out that some of the points addressed here are also found in textbooks these days. But I see this manuscript not as a research paper, but a broad, "high-altitude" view of the subject, and in that context these points have their place. I concur with many of the points raised by Gerard Govers, and commend them

to the author for his consideration. I would suggest that the manuscript be published without further review, after the author makes any changes that he sees fit in response to the comments received.

---

## Author Comment (AC1) · 23 Nov 2016

Summary of responses to referees – to be read with tracked changes revision of ms.(attached)

Referee 1: I have addressed almost all of the points raised. I have taken 2 paragraphs out of sections 1-3, but am reluctant to remove more, as that sets the flavour of the whole piece. I have reworked the section around figure 6 and near line 550 (see under referee 2 below) Added references are Ali et al 2013 Critchley et al 1991 Jacobson et al, 2000 Steffen et al, 2015.

Referee 2 (Govers): I have adopted all suggestions. The most substantial change is in re-shaping the discussion around line 550. Text has been modified to reflect yields and relevance of poor agronomic practice, and figure 6 (and its caption) has been replaced.

A number of additional references have also been included, namely: Attal et al, 2015 Barthold & Woods, 2015 Davidson, 2009 Dere et al, 2010 Herold et al, 2014 Janzen& McDonnell, 2015 Johnson et al, 2014 Kirchner et al, 2000 Knapen et al, 2007 Larsen et al, 2012 Sadras et al, 2011 Stauffer & Aharony, 1985 Vitousek et al, 2016

Referee 3: No specific corrections

Please also note the supplement to this comment:
http://www.soil-discuss.net/soil-2016-50/soil-2016-50-AC1-supplement.pdf

**Supplement:**

[revised manuscript text omitted]